# Chasing the Youth Dividend in Nigeria, Malawi and South Africa: What Is the Role of Poverty in Determining the Health and Health Seeking Behaviour of Young Women?

**DOI:** 10.3390/ijerph192114189

**Published:** 2022-10-30

**Authors:** Sibusiso Mkwananzi, Ololade Julius Baruwa

**Affiliations:** 1Institute for Gender Studies, College of Human Sciences, University of South Africa, Pretoria 0002, South Africa; 2Centre for Demographic Research, Catholic University of Louvain (UCL), 1348 Louvain-la-Neuve, Belgium

**Keywords:** demographic dividend, poverty, reproductive health, HIV, health seeking behaviour

## Abstract

Africa’s new source of hope lies in harnessing Demographic Dividend, which may create a window of economic opportunity as fertility levels decrease if the correct policies and programmes are put in place. It has been shown that the health status of young people should be optimal for the realisation of the demographic dividend. This study examined the association between poverty and the health status and health seeking behaviour of young women (15–25), using the Demographic Health surveys of Nigeria 2013, Malawi 2015–2016, and South Africa 2015–2016. Interest variables were household and community-levels of poverty, while the outcomes were pregnancy and HIV testing and health services seeking. Results showed that in Malawi only household poverty was associated with a higher likelihood of pregnancy, while in South Africa household- and community-levels of poverty were associated with a higher likelihood of pregnancy. In Nigeria, household- and community-levels of poverty were not significantly linked to pregnancy but were associated with a lower likelihood of HIV testing and health seeking behaviour. The study shows empirical evidence of the construction of negative health outcomes in poor households and communities in Nigeria, Malawi, and South Africa. Therefore, if the demographic dividend is to be a reality in the near future, it is imperative to ensure that poverty-alleviation urgently occurs.

## 1. Introduction

Africa’s new source of hope appears to lie in the harnessing of the Demographic Dividend (DD), a window of economic opportunity that may occur as fertility levels decrease nationally if policies to ensure economic development are in place. The demographic dividend has been cited as one of the predominant reasons for the highest level of economic development ever seen globally that occurred in East Asia [1]. The process of acquiring benefits from the demographic dividend is unfortunately not automatic [2]. Countries that go through demographic transition may indeed see youth bulges and a large proportion of working-aged individuals, but if the appropriate policies are not in place to ensure that these individuals are adequately employed, economic development will not occur [3]. It is estimated that 41% of the population worldwide lives in poverty, while children that grow up in these households face mental, physical, and social challenges, as well as higher mortality due to higher levels of malnutrition and childhood illness [4]. Thus, all policies need to be pro-poor, thereby addressing the hardships faced by the most vulnerable members of society. To do this, governments should consistently keep the needs, barriers and challenges faced by the poor in mind and respond to these appropriately, sensitively and to the convenience of this group. Moultrie (2007) [5] highlighted that for the dividend to produce economic growth in a nation, the majority of the working age population has to be employed or at least nearing employment. If this was not ensured, each year would result in lower amounts of productivity, higher levels of delinquency, increasing crime rates, violence and public protests instigating political instability [2,5].

Globally, the median age in 2018 was 30 years old [6]. In 2019, there were 1.2 billion youth aged 15–24 years globally [7]. In Africa, approximately 60% of the population is under the age of 25, which is about 800 million individuals, and the continent is set to have the largest population growth rate in the future globally as these young people enter their reproductive stage [8,9]. General awareness of the benefits obtained from the youth bulge and how youth can contribute towards development exists. The expectation of the youth bulge is that it has the potential to propel and stimulate economic growth through the rise of a youthful population compared to other age groups. However, this is not the case in sub-Saharan Africa, where a high rate of risky behaviours among the youth has posed as a problem rather than a profit to the society at large [10,11,12]. In response, governments have attempted several strategies, such as empowerment programmes to address the challenges facing youth to harness their full potential, improving themselves and national economies. For instance, the National Youth Policy (NYP) in South Africa and Malawi, as well as the Nigeria Youth Employment Action Plan, were all implemented to consolidate and integrate youth into the mainstream of government policies for economic reconstruction and recovery [13,14,15]. Thus far, the demographic dividend has not been reaped, despite the window of opportunity opening in 2000 [2,5]. Sub-Saharan Africa can still benefit from the demographic dividend if policy is geared toward decreasing fertility levels, as well as regional conflicts, educating and empowering women, decreasing the morbidity and mortality levels among the working-age population and finally substantially improving the quality of the economy and solid institutions (political systems, political freedom, rule of law, corruption, infrastructure and trade openness) on the subcontinent [2,9,16].

It is evident that there is a youth bulge in Africa, and as the region with the youngest population in the world, improving health outcomes, especially among youth, remains essential. There are several studies that have shown that there is strong evidence linking poverty and health status [17,18,19], but these studies are old. More recent studies have developed and increased a focus on the link between income and the health of populations [20,21,22]. Khullar et al. [20] found that income is negative and strongly associated with mortality and morbidity because individuals with low incomes face great barriers in accessing medical care. Income level could also influence the health status and health seeking behaviour of an individual through standards of living, nutrition, mental stress, and educational level, among others, making people in higher-income groups have better health status [22]).

Additionally, the health status of young people should be optimal for the realisation and harnessing of the demographic dividend [2,5,7]. Evidence has shown that a healthy population can facilitate economic growth and lessen poverty [2,23]. Berry and Kim [23] showed that with increasing life expectancy, development levels accelerate up to a maximum of 72 years among males and 75 years among females. Also, a healthier population would mean greater physical capital, productivity, and enhanced performance in educational endeavours, furthering economic growth of a nation [23]. Therefore, this study aimed to examine the association between poverty and the health status of young women (15–25) in Nigeria, Malawi and South Africa. Specifically, the study sought to answer the research questions: What is the level of health status outcomes among young women in Nigeria, Malawi, and South Africa? Also, what is the association between poverty and health outcomes of young women in Nigeria, Malawi, and South Africa? Nigeria, Malawi and South Africa therefore become important settings for this study because of their youth bulge. These countries were chosen due to their similarities despite varied locations. Specifically, Malawi is similar to South Africa in terms of the effects of the HIV pandemic, while Nigeria is similar to South Africa with regards to its economy.

## 2. Methods

This quantitative analytical study is based on the secondary analysis of existing nationally representative secondary data of the Demographic and Health Surveys (DHS), from three countries: Nigeria (2013), Malawi (2015–2016) and South Africa (2015–2016) [24]. The study made use of the most recent datasets that had all the outcome variables present. Although 2013 Nigeria DHS is not the most recent DHS for Nigeria, the 2018 Nigerian DHS does not have the HIV testing variable. Therefore, these datasets were used to ensure uniformity in variables and because this study is trying to compare the findings across a period nearest to 2015. Moreover, the individuals of 2013 and 2015 may be comparable due to the cross-sectional snapshots that were taken around the same time.

The study sample consisted of 15–25-year-old females. The analysis included 11,243 young women from Malawi, 16,946 female participants from Nigeria and 3176 young women from South Africa. The outcomes of the study were pregnancy, HIV testing and health seeking behaviour. The pregnancy outcome in this study was measured as yes and no, indicating the current pregnancy status of young women at the time of the survey. The HIV testing outcome was measured as yes and no, indicating whether or not young women have ever tested for HIV, a variable suggesting attitude towards health status. Lastly, the outcome variable “health seeking behaviour” was measured as yes and no, indicating the utilization of health facilities by young women in the last 12 months.

Some of the background control factors included in the study were operationalized and reconstructed for the analysis. Age was measured as a continuous variable; marital status was measured as “never married and ever married”; place of residence was measured as “urban and rural”; education attainment was measured as “none, primary, and secondary/tertiary”; and employment status was measured as “employed and unemployed”. The interest variables were household poverty (HH poverty), measured as “no HH poverty, HH poorer, HH poorest” using the household wealth index variable while the community-levels of poverty were measured as a percentage using the aggregated household poverty status from the wealth index variable.

STATA 15 (StataCorp, College Station, TX, USA) was used for the data analysis. Results were interpreted using odds ratio, while the level of significance was set at 5%. Data analysis incorporated descriptive analysis through the percentage distributions of categorical variables, as well as the median and inter-quartile ranges for numerical variables. An inferential statistical analysis to determine the association of poverty and health involved multivariate binary logistic regression, controlling for all variables, and reporting adjusted odds ratios.

## 3. Results

The characteristics of the sample included in this study are shown in Table 1.

Table 1 displays the distribution of the characteristics of the study participants. The median age across the three countries was 20-years-old with an inter-quartile range of 17 to 23 years in Nigeria and South Africa, while Malawi had an inter-quartile range of 17 to 22-years-old. The greatest proportion of young women in Malawi and Nigeria were married, employed and living in rural areas. In South Africa, most young women were living in urban areas, never married, unemployed and had secondary education. Approximately two in five girls were living in poor households across all three countries. Regarding health indicators, about a tenth of girls were currently pregnant in Malawi and Nigeria, while only 4% were pregnant in South Africa. Finally, most young women had tested for HIV and visited a health facility in Malawi and South Africa, but this was not the case in Nigeria.

The results of the multivariate analysis are presented in Table 2. Inferential results showed that in Malawi poverty of the household was positively associated with the likelihood of pregnancy only with young women from the poorest households having 38% higher odds of pregnancy and those from poorer households having 25% increased likelihood of pregnancy. In Nigeria household and community poverty were not associated with pregnancy. However, both household and community poverty decreased the likelihood of HIV testing, as well as seeking health services. For instance, young women in the poorest household in Nigeria were 0.28 times less likely to go for HIV testing. Similarly, community poverty in Nigeria reduces the risk of seeking health services (OR = 0.53; *P*-value = 0.00). Finally, in South Africa, household and community poverty were not significantly associated with health seeking behaviour. However, both household and community poverty increased the likelihood of pregnancy, while young women living in poor households had increased odds of HIV testing.

## 4. Discussion

The first aim of this study was to examine the level of health and health seeking behaviour among young women in Malawi, Nigeria, and South Africa. The sample for the current study differs by age (15–25 years) compared to most previous studies, which makes our study unique in terms of population sample. Hence, findings from this study might be a little bit difficult to compare with existing studies. However, the study age range was chosen due to reproductive health factors being collected from young women aged 15 years to 49 years in the DHS. Pregnancy amongst 15–25 year olds could still be considered amongst young women, as most are either attending tertiary education or have just begun working.

This study found that 10% of young women in Malawi, 13% in Nigeria and 5% of female youth in South Africa were currently pregnant. The level of women that were found to be currently pregnant in Malawi is quite different from a previous study conducted in Malawi that showed levels of pregnancy among unmarried teenagers to be 27% [25]. With respect to pregnancy among young women in South Africa, a recent study reported that between 2017 and 2021, the number of births to young teenagers aged 10–14 years increased by 48%, and the number of births to teenagers 15–19 years increased by 17.9% [26]. Previous studies in Nigeria have shown that pregnancy among young women remains a problem [27,28]. Adolescent pregnancy rates remained unchanged in Nigeria between 2008 and 2013 at 22.9% and 22.5%, respectively [28].

Results from this study show that Malawi had the highest levels of HIV testing at 71%, followed by South Africa at 70% and 21% among young women in Nigeria. The level of HIV testing among young women in Malawi and South Africa is encouraging compared to that of Nigeria. However, the above findings show that full HIV testing uptake among young women who are among the most at-risk population is yet to be achieved, especially in Nigeria. The finding that HIV testing is low among young women (15–25 years) in Nigeria shows a slight decrease in the uptake of HIV testing in this age group compared to the finding of a previous study conducted among females between the ages of 15 and 24 years in Nigeria, which reported HIV testing at 19% in 2013 [29]. The low percentage of HIV testing among young women in Nigeria has been said to be due to the absence of knowledge of HIV, low perceptions of personal risk, low access and the high cost of testing and stigma [30]. Therefore, there is a need to design and implement strategies tailored to improve knowledge of HIV, expand access, and eradicate stigma associated with HIV testing uptake.

This study revealed that health seeking behaviour is fairly better among young women in South Africa and Malawi as opposed to women in Nigeria. A study on factors influencing health seeking behaviour among civil servants in Ibadan, Nigeria reported much higher health seeking behaviour among study participants at 86.5% compared to our study’s results of 18% of young women in Nigeria having visited health facilities in the last 12 months [31]. While a previous study in South Africa, which revealed the level of health seeking behaviour to be 53% among 18-year-olds and older, using the General Household data is similar to the findings of this study, which found that 57% of young women in South Africa and 55% in Malawi sought health care services [32]. The results of this study show that the willingness to seek health care services among young women when the need arises is a matter of concern in the three countries, especially in Nigeria, where the level of health seeking behaviour is lowest.

The second objective of the study was to analyse the association between poverty and health outcomes of young women in Malawi, Nigeria, and South Africa. The study revealed that in Malawi and in South Africa, HH poverty is associated with a higher likelihood of pregnancy among young women. In addition, the results from this study also showed that community level poverty is associated with a higher likelihood of pregnancy among young women in South Africa. Several studies have demonstrated the effects of poverty on teenage pregnancy [33,34,35]. The finding that HH poverty is associated with pregnancy among young women is consistent with a previous study in Malawi, which revealed that wealth-related inequality contributes largely to teenage pregnancy and childbearing in Malawi, while teenage pregnancy and childbearing worsen among young women from poor households [33]. Another previous study in South Africa, which is in line with the findings of the current study, revealed that though there are other factors that contribute to pregnancy, poverty was the major reason why young women fall pregnant [36].

Poverty has a bi-directional relationship with pregnancy outcomes among women, where poverty can be a determinant of pregnancy, and a consequence of pregnancy, especially among young women. Poverty creates a lack of resources and support that limits educational and financial opportunities that may reduce the risk of pregnancy among women. Moreover, it has been established that poverty may serve as a pre-existing condition that affects pregnancy; reasons for this are attributed to early marriage, poor knowledge, unavailability and low use of contraceptives and poor economic infrastructure, which are consequences of poverty [37].

This study further demonstrates that there is a strong association between poverty and HIV testing. In Nigeria, HH and community poverty are associated with lower HIV testing, while HH poverty was found to be associated with higher HIV testing in South Africa. The finding that poverty is associated with low HIV testing corroborates the findings of [38] in Nigeria, which revealed that the likelihood of HIV testing is doubled among adolescents and young adults from wealthy households compared to those in poor households. A possible explanation for lower HIV testing among those from poor households or community could be the low socioeconomic capacity to access health services. Contradictory to the findings of Jooste et al., (2021) [39], the finding from this study showed that HIV testing is higher among poor young women compared to those in the rich households of South Africa. However, the result that HH poverty is associated with higher HIV testing in South Africa can be linked to incentives given to people who volunteered to test for HIV such as money, clothes, headsets and flash drives, among others. Moreover, studies have established a positive relationship between incentives or rewards and HIV testing in South Africa [40,41,42].

The finding that HH and community poverty is associated with low health seeking behaviour in Nigeria is consistent with previous studies from Nigeria (Atchessi et al., 2018; Latunji and Akinyemi, 2018) [31,43]. The inability to pay for health services is perhaps a major reason for low health seeking behaviour among individuals from poor communities or households [44].

The implication of the association between poverty and pregnancy to the demographic dividend is that childbearing may undermine the health of young women and their socioeconomic prospects. Xu and Shtarkshall (2004) [45] showed that teenage pregnancy prevention has benefits for a society in general. Delaying pregnancy beyond the teenage years has been found to correlate with higher female educational attainment. This in turn is passed on in raised levels of community knowledge, skills, employment prospects and the chance of a productive life [46,47,48]. Higher rates of secondary school completion have been found to correlate with lower fertility in the teenage years and subsequently lower population growth rates, which aid in the development of a country [45]. Consequently, teenage pregnancy could contribute to the delayed or non-acquisition of the demographic dividend. Improving human development by reducing pregnancy among young women can reduce fertility rates, which in turn can play a key role in boosting per capital income and reducing poverty. This can be achieved by following some of the World Bank policy guidelines to facilitate fertility transition, expanding education without letting girls fall behind, and empowering women, as well as increasing access to comprehensive family planning services [49].

The implication of the association between poverty and health seeking behaviour is that poverty influences health status at every stage of life, which may have a multiplicative effect on unhealthy existences [50]. Poverty is a general condition restricting people’s ability to prevent or seek treatments for illness, which can create threats to the living conditions and health status of people [51]. Another angle to explain the effect of the association of poverty and health seeking behaviour on poverty could be that the upstream social determinant of health (such as poor education, lack of access to health care and housing, etc.) caused by poverty can compromise the social structure that influences the health status of an individual, while ill-health caused by poverty will reduce productivity among people.

This study has provided empirical evidence of the construction of negative health outcomes among young women residing in poor households and communities in Nigeria, Malawi and South Africa. Despite the findings and contribution of this study to reproductive health and demographic dividend literature, some important points should be considered while interpreting the findings of this study. First, because the study used a cross-sectional survey, the findings from this article are limited to statistical associations; thus, potential causal relationships of variables cannot be ascertained. Second, based on the cross-sectional nature of the datasets used, the data potentially suffer from both recall and reporting bias.

## 5. Conclusions

While the youth bulge presents great potential for economic opportunities through active participation in economic activities, our study confirmed that poverty is detrimental to achieving demographic dividend in our study population. This is because poverty is linked with pregnancy and low health seeking behaviour among young people. Therefore, if the demographic dividend is to be a reality in the near future within these countries, it is imperative to ensure that poverty-alleviation occurs urgently. It is recommended that pro-poor policies and programmes that aim to increase skills development and employability be implemented to decrease adverse health outcomes among young women, as well as drive economic development.

## Figures and Tables

**Table 1 ijerph-19-14189-t001:** The distribution of characteristics of the study participants.

Characteristics	Malawi (*n* = 11,243)	Nigeria (*n* = 16,946)	South Africa (*n* = 3176)
Age (median; IQR)	20; 17–22	20; 17–23	20; 17–23
Ever Married	47.36%	47.08%	4.06%
Education	62.65%—Primary	50.48%—Secondary (30.08%—None)	86.46%—Secondary
Employed	50.69%	41.10%	11.39%
Rural residence	78.23%	61.15%	45.94%
Currently pregnant	10.14%	12.73%	4.62%
Ever Tested for HIV	71.41%	21. 04%	70.07%
Visited Health Facility	55.39%	17.59%	56.69%
HH Poverty	37.25%	36.26%	45.65%

**Table 2 ijerph-19-14189-t002:** The adjusted association between poverty and health outcomes among young women in Malawi, Nigeria and South Africa.

Characteristic	Malawi	Nigeria	South Africa
**Pregnancy**			
HH Poorest	1.38 (0.02) **	1.06 (0.64)	2.43 (0.04) **
HH Poorer	1.25 (0.09) *	1.10 (0.45)	2.25 (0.06) *
Comm Poverty	1.08 (0.33)	0.93 (0.27)	1.53 (0.04) **
**Ever HIV Testing**			
HH Poorest	0.86(0.14)	0.28 (0.00) **	2.33 (0.00) **
HH Poorer	0.89(0.27)	0.48 (0.00) **	2.56 (0.00) **
Comm Poverty	0.91 (0.15)	0.54 (0.00) **	1.13 (0.25)
**Health Seeking**			
HH Poorest	1.02(0.78)	0.42(0.00) **	1.18 (0.29)
HH Poorer	1.09(0.29)	0.67(0.00) **	1.20 (0.24)
Comm Poverty	0.94 (0.18)	0.53 (0.00) **	1.05 (0.57)

** indicates *p*-value less than 0.05; * indicates *p*-value less than 0.10; *p*-value is indicated in the parenthesis.

## Data Availability

This research made used of DHS datasets, which is publicly available. However, permission to use the datasets was granted by dhsprogram. Detailed information regarding procedures and questionnaires are reported on DHS website https://dhsprogram.com/ (accessed on 7 March 2022).

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
