# Peer review of "Chasing the Youth Dividend in Nigeria, Malawi and South Africa: What Is the Role of Poverty in Determining the Health and Health Seeking Behaviour of Young Women?"

_ijerph, 2022, doi:10.3390/ijerph192114189_

Round 1

Author Response

Addressing reviewer’s comment

Reviewer’s Comments

How it has been addressed

Title

·        Rephrase the title to be more concise

·        Chasing the Youth Dividend: What is the Role of Poverty in Determining the Health and Health Seeking Behaviour of Young Women?

Abstract

·        Lines 10-11 “a window of economic opportunity that occurs as fertility levels decrease” should be rephrased with a nondebatable statement

·        Replace DHS 2015 with the most recent DHS round

·        Africa’s new source of hope lies in harnessing Demographic Dividend, which may create a window of economic opportunity as fertility levels decrease if the correct policies and programmes are put in place.

·        This study examined the association between poverty and the health status and health seeking behaviour of young women (15-25), using the Demographic Health surveys of Nigeria 2013, Malawi 2015-16, and South Africa 2015-16.

·        The 2018 Nigerian DHS does not have the HIV testing variable of interest

Introduction

·        Line 38-39: statement requires referencing

·        Line 86: replace with “… negative and strongly associated with.”

·        Deleted

·        that income is negative and strongly associated

Methods

·        Line 86: replace with “Nigeria (2018), Malawi (2015-16) and South Africa (2016). The three reports should be referenced

·        How were the outcome variables measured? Do the authors mean to say HIV test and health seeking behaviours? There is a need for clear operation definition of outcome variables that will align with the title of the study.

·        Justify the choice of Malawi, Nigeria, and South Africa

·        Line 115: replace with the statement “STATA 15 was used for data cleaning, storage and all analysis” with “STATA 15 was used for data analysis at 5% level of significance”

·        MEASURE DHS [Internet]. Demographic and Health Surveys. Calverton: ICF International; 2022. Available from: http://www.measuredhs.com/ [accessed 21 October 2022]

·        The pregnancy outcome in this study was measured as ‘yes’ and ‘no,’ indicating the current pregnancy status of young women as at the time of the survey. The HIV testing outcome was measured as ‘yes’ and ‘no,’ indicating if young women have ever tested for HIV, a variable suggesting attitude towards health status. Lastly, the outcome variable health seeking behaviour was measured as ‘yes’ and ‘no,’ indicating the utilization of health facilities by young women in the last 12 months.

·        Nigeria, Malawi and South African become an important setting for this study because of their youth bulge

·        STATA 15 was used for data analysis. Results were interpreted using odds ratio while level of significance was set at 5%.

Results

·        Line 121: change on to in to read as “… included in this study…)

Table 1

·        The table could have sufficed if there is a clear description of the variables included in the study. For instance, how were the HH and community poverty level measured and categorised? Thus, authors should indicate each variable’s categories.

·        Why are the community poverty corresponding row values empty?

Table 2:

·        Authors should indicate what asterisks (* and **) represent as a footnote to Table 2

·        Also, indicate what the values in the parenthesis stand for. For instance, if it is the p-value let it be mentioned as such as a footnote

·        Corrected

·        Some of the background control factors included in the study were operationalized and reconstructed for the analysis. Age was measured as a continuous variable; marital status was measured as ‘never married and ever married’; place of residence was measured as ‘urban and rural’; education attainment was measured as none, primary, and secondary/tertiary’; and employment status was measured as ‘employed and unemployed.’ The interest variables were household poverty (HH poverty), measured as ‘no HH Poverty, HH Poorer and HH Poorest’ using the household wealth index variable while the community-levels of poverty were measured through obtaining the percentage of poorer and poorest households within the community from the aggregated household wealth index variable.

·        Corrected through removing this variable from the table completely.

·        Corrected through explanation being added under the table

·        ** indicates p-value less than 0.05; * indicates p-value less than 0.10

Discussion

 Paragraph 1:

·         I suggest authors should expound on the age differences in the reported literature compared to their study as a possible cause of the variations in the reported prevalence.

·        Again, most of these reported studies either focused on adolescents or teenage pregnancy, not only in this paragraph but also throughout the discussion!

·         Page 6, lines 230-246 could have been better appreciated if summarized and mentioned in the introduction section. Nonetheless, the concern remains the age range (15-25) in this study, focusing on young women but neither adolescent nor teenager

·        The sample for the current study differs by age (15-25 years) compared to most previous studies, which make our study unique in terms of population sample. Hence, findings from this study might be a little bit difficult to compare with existing studies.

·        The DHS collects reproductive-related information from young women from the age of 15 to 49 years only. Hence, the study looked at youth/young women aged 15-25 years of age. This still represents pregnancy amongst young women. We have specifically not referred to this as teenage or adolescent pregnancy due to the knowledge that 20–25-year-olds are included in the study.

·        The DHS collects reproductive-related information from young women from the age of 15 to 49 years only. Hence, the study looked at youth/young women aged 15-25 years of age. This still represents pregnancy amongst young women.

Reviewer 2 Report

Excellent topic of the article and although there is use of secondary data, the cut had quantitative force. In this cross-sectional study, the only caveat in the methodology would be to place the value that the researchers consider significant, I think it was p<0.1 because there was a significance level of p=0.09. I think the text is succinct and direct in its discussions and can be published.

Author Response

This is corrected through explanations being added under the table

** indicates p-value less than 0.05; * indicates p-value less than 0.10